# Miraculous Al/PDF Composites Using NF_2_ to Enhance the Energy Release of Al, Prepared Through an Efficient Method

**DOI:** 10.3390/nano14241980

**Published:** 2024-12-10

**Authors:** Junqi He, Jing Lv, Wenfang Zheng, Renming Pan, Yanan Li

**Affiliations:** 1School of Safety Science and Engineering (School of Emergency Management), Nanjing University of Science and Technology, Nanjing 210094, China; hejunqi66@sina.com (J.H.); panrenming@njust.edu.cn (R.P.); liyanankyzy@yeah.net (Y.L.); 2School of Chemistry and Chemical Engineering, Nanjing University of Science and Technology, Nanjing 210094, China; lvjing9487@163.com

**Keywords:** NF_2_, Al powder, PDF, composites, coating, energy release

## Abstract

To enhance the energy release of Al powder in solid propellant, ploy (difluoroaminomethyl-3-methylethoxybutane) (PDF), which has difluoroamino (NF_2_), was utilized to improve energy and promote combustion efficiency. In this study, Al with three distinct powder sizes (29 μm, 13 μm, and 1~3 μm) was coated with PDF using the solvent/non-solvent method, leading to the formation of Al/PDF composites. The morphology and characteristics of Al/PDF were then characterized. The results demonstrated that all powder sizes of Al/PDF had core-shell structures, and NF_2_ of the PDF layer on the Al surface maintained the original structure. The TG curves indicated the amount of the PDF layer related to the powder sizes. Furthermore, Al/PDF exhibited greater hydrophobicity. NF_2_ prompted Al/PDF, with better catalysis on ammonium perchlorate (AP) decomposition. Compared to Al powder, the ignition delay of Al/PDF was significantly shortened. For mixed samples of Al/PDF and AP, NF_2_ shorted the ignition delay, improving combustion stability, extending the combustion duration, and forming volatile fluorine compounds. These findings underscore the effects of NF_2_ in Al/PDF composites, which enhances the energy release of Al and holds promising potential applications.

## 1. Introduction

Aluminium (Al) powder is the ubiquitous metal fuel used in solid propellants, and its practical energy release significantly impacts the performance of solid propellants [1,2,3]. The enhanced energy release is a consequence of the increased energy in Al powder and its improved combustion efficiency [4,5]. Based on the above, enhanced Al energy release, especially combustion efficiency, has garnered significant research interest [6,7,8,9,10]. Of these, it is feasible to address the objectives mentioned above through Al and the special functional materials of combined composites. Fluorinated materials [11,12,13,14] are incorporated with Al as composites, which are in the form of fluorinated polymers such as perfluoro tetra decanoic acid (PFTD), polyvinylidene fluoride (PVDF), or polytetrafluoroethylene (PTFE) [15,16,17,18,19]. These make AlF_3_, formed during the solid propellant reaction, easier to decompose and volatilize, and during the decomposition, it is highly reactive with Al and Al_2_O_3_, allowing a pre-ignition reaction (PIR); thus, the fluorinated materials improve the combustion efficiency of the Al powder. Regrettably, however, these fluorinated materials are not energetic, resulting in a possible reduction of Al’s energy release. Meanwhile, compounds like nitrocellulose (NC), ammonium perchlorate (AP), and glycidyl azide polymer (GAP) [20,21] can introduce energetic groups for the Al powder, increasing the energy [22,23]. This enhancement of the energy of Al powder is unable to improve its combustion efficiency but is equally significant for solid propellants. Nevertheless, considering the performance of solid propellants, it is preferable to combine Al powders with energetic fluorinated materials, which have all the functions of the two methods mentioned above. However, research on concise and effective methods to incorporate target groups on the surface of Al powder to meet the above requirements is still in its infancy.

Difluoroamino (NF_2_) has a theoretical density of 2.303 g/cm^3^ and an enthalpy of formation of −32.7 kJ/mol [24]. The N-F bond has higher reactivity than the C-F bond, with the F in NF_2_ appearing in the form of the oxidant. When NF_2_ is introduced into the structure of the compound, the energy level of the compound can be significantly enhanced and, thus, can be used as a potential oxidant or binder in solid propellants [24,25]. Notably, the HF produced by NF_2_ during decomposition has a low relative molecular mass and a bond energy as high as 565 kJ/mol. The released energy of Al and F during the reaction is approximately 56.10 kJ/g, which is approximately twice that (30.98 kJ/g) of Al and oxygen [26,27]. At the same time, the F in NF_2_ is a good combustion aid in improving the combustion efficiency of Al powders, which is as effective as F in fluorinated polymers [27,28]. Therefore, NF2 combines the functions and advantages of the F elemental group and the energy group, which provides a good breakthrough for solving the above problems. The emergence of NF_2_ offers a novel solution to the issues mentioned above if incorporated into the Al powder and reacts with Al. Poly (difluoroaminomethyl-3-methylethoxybutane) (PDF) [27] is a kind of difluoroamino polymer with an active telohydroxyl group (-OH) and can facilitate interaction with the Al powder; thus, PDF serves as an NF_2_ material and is combined with Al powder to synthesize Al/PDF composites.

The oxide layer on the surface of Al powder protects and isolates the external chemical reaction, thus hindering the direct bonding of PDF with the Al surface. It must be removed as much as possible through a low-concentration alkali solution. At the same time, the alkali solution is able to hydroxylate the Al surface, making PDF easier to bond with Al. NF_2_ with strong reactive can quickly occur reaction, and it is necessary to prevent NF_2_ from other reactions that may lead to its structural destruction, thus rendering NF_2_ useless. According to previous studies and our previous work on the similar metal fuel boron [29], the coating is a common and efficient method for functional materials added for Al, which can enhance the specific surface area of Al powder and improve its surface reactivity with no drastic change in morphology [10,23]. In this study, Al/PDF composites were prepared through surfactants and the solvent/non-solvent method [2], which can react rapidly and efficiently, and isocyanate was used with PDF linked with the Al powder. We successfully incorporated NF_2_ of PDF on Al powder with three different powder sizes (29 μm, 13 μm, and 1~3 μm), and Al/PDF was synthesized. Furthermore, the structure, morphology, and hydrophobicity of Al/PDF were investigated. It has been confirmed that NF_2_ of PDF significantly improves the total performance of Al powder. More importantly, this verifies the facilitating effect of NF_2_ on the ignition of Al/PDF and the catalytic behavior of Al/PDF on AP.

## 2. Experimental Section

### 2.1. Materials

Aluminium (Al) powders with sizes of approximately 29 μm, 13 μm, and 1~3 μm were acquired from the Shanghai Academy of Spaceflight Technology (Shanghai, China). 95% ethanol, ethyl acetate, and hexane, all of analytical purity, were brought from Wohua Chemical Co., Ltd. (Shanghai, China). Sodium hydroxide (NaOH) and (3-aminopropyl) triethoxysilane (KH-550), all of the analytical grades, were provided by Sinopharm Chemical Reagent Co., Ltd., (Shanghai, China). Analytically-pure methylidynetri-p-phenylene tri-isocyanate (20% in C_6_H_5_Cl) was purchased from Macklin (Shanghai, China). Ammonium perchlorate (AP) with a particle size of approximately 150 μm was purchased by the Xi’an Modern Chemistry Research Institute. PDF was synthesized using Li’s et al. [27] method. The molecular mass (*M_n_*) of PDF was 3600 g/mol, with a polydispersity index (*M_w_*/*M_n_*) of 1.6.

### 2.2. Pretreatment of Al Powder

The pretreatment of Al powder involves adding the Al powder of each powder size (29 μm, 13 μm, 1~3 μm) to 0.01 mol/l of NaOH solution to remove the oxide layer. The above-treated Al powder was thoroughly dispersed in ethanol and then KH-550 was added. After ultrasonic dispersion for 1 h, the suspension of Al powder was magnetically stirred at 60 °C for 2 h under nitrogen (N_2_). At the end of the reaction, the Al powder was filtered and dried in a vacuum oven at 60 °C for 12 h.

### 2.3. Preparation of Al/PDF Composites

In this study, we prepared Al/PDF with various powder sizes using the solvent/non-solvent method. A schematic of the preparation process is shown in Figure 1. In a typical step, 2 g of pretreated Al powder was dispersed in 50 mL of ethyl acetate, followed by the addition of 10% wt PDF, and then ultrasonic dispersion for 30 min. The mixture was then slowly added to 250 mL of hexane containing methylidynetri-p-phenylene tri-isocyanate with vigorous magnetic stirring under N. It is important to note that the temperature of the hexane was 60 °C, and the mixture was added at a rate of 1 mL/min. After the addition of the mixture, the temperature was reduced to room temperature (20 °C). The precipitated PDF adhered to the surface of the Al powder and formed the desired coating with the help of the methylidynetri-p-phenylene tri-isocyanate. After the addition of the mixture, the reaction was carried out for 2 h at room temperature. Subsequently, the coated Al powders were filtered, washed twice with hexane, and dried under vacuum at 60 °C. It is worth noting that the coating technique was similar for each powder size.

### 2.4. Characterizations and Measurements

The surface chemical structure of Al/PDF was measured by Fourier transform infrared (FTIR) (NICOLETIS 20, Thermo Fisher Scientific, Waltham, MA, USA) with a range of 4000 to 500 cm^−1^. The surface morphology of Al/PDF was observed by field emission scanning electron microscope (FESEM) (FEI QUANTA 400 FEG, Oxford Instruments, Abingdon, Oxfordshire, UK) and transmission electron microscope (TEM) (JEM-2100F, Akishima, Tokyo, Japan). The X-ray diffraction (XRD) patterns of the samples were conducted on Smartlab 9 (Rigaku, Akishima-shi, Tokyo, Japan) (2*θ* = 10~90°). Additionally, the elemental state of Al/PDF was conducted by X-ray photoelectron spectroscopy (XPS) (ESCALAB 250Xi, Thermo Fisher Scientific, Waltham, MA, USA).

The static contact angle of the samples (Al or Al/PDF) was measured using a surface tensiometer (SDC-100, Dongguan, China). A hydraulic machine was used to press each sample (500 mg) into a wafer (10 mm in diameter) at 10 MPa, then the pressed sample was treated on the test bed of the surface tensiometer in turn, using 4 μL of deionized water drops each test. Each sample was tested five times, and the average was taken as the reference value. Thermogravimetric analysis (TGA) (SDTA851E, Mettler Toledo, Zurich, Switzerland) and differential scanning calorimetry (DSC) (DSC3, Mettler Toledo, Zurich, Switzerland) with the heating rate of 10 K/min, and a temperature range of 50~500 °C under N_2_ atmosphere. TGA was used to examine the thermal mass loss of Al or Al/PDF (weighed 0.6~0.8 mg), and DSC was tested the exothermic characteristics of the mixed sample of Al and AP, Al/PDF, and AP. The mass ratio of DSC samples (AP:Al or Al/PDF) was 9: 1, and weighed 0.6~0.8 mg. Pyrolysis–gas chromatography/mass spectrometry (PY-GC/MS) analyzed volatile compounds of mixed samples (AP and Al/PDF), and the pyrolysis temperatures were set as 300 °C and 500 °C, respectively. The mass ratio of AP:Al/PDF was 9:1, and around 0.2 mg samples were placed in the Frontier 3030D pyrolyzer interfaced with a GC/MS (QP2010ULTTA, Shimadzu, Kyoto, Japan). The relevant parameters were set as follows: He as the carrier gas, pyrolysis and GC/MS interfaces were kept at 300 °C, the GC was temperature-programmed from 40 °C (for 2 min) to 140 °C at 10 K/min, and from 140 °C to 300 °C at 20 K/min, and the electron impact mode (70 eV) for m/z 10~550.

For Al or Al/PDF laser ignition and combustion, each sample (Al or Al/PDF) was weighed to 20 mg and then put into the quartz tube (diameter: 3 mm, length: 5 mm). For mixed samples laser ignition and combustion, each sample was mixed with a ratio of 65: 35 (AP: Al or Al/PDF) [29,30], weighed to 20 mg, and put into the quartz tube (diameter: 3 mm, length: 5 mm). At the end of the combustion of the mixed samples, the residues were collected and then characterized by FESEM using the above method. A high-speed camera recording (Photron, FASTTXAM Mini UX100, Tokyo, Japan) performed at 1000 fps was employed to detect the above ignition and combustion process. The parameters of the laser ignition device provided by Nanjing University of Science and Technology were set as voltage: 220 V ± 10%; laser power: 40 W; laser spot: 5 mm; light duration: 1 s [30].

## 3. Result and Discussion

### 3.1. Structure and Morphology of Al/PDF

Al/PDF with powder sizes of 29 μm, 13 μm, and 1~3 μm are shown in Figure 1, where spectra are compared with the Al without coating and PDF layer. The FTIR spectra revealed significant changes in the Al/PDF with different powder sizes compared to the uncoated Al powder. Moreover, the intensity of the characteristic peaks of the Al/PDF weakened as the powder size decreased. Notably, Al/PDF with a size of 29 μm exhibited the most apparent characteristic peaks (Figure 1a,b). Generally, the characteristic peaks of all Al/PDF were identified. The N-H stretching vibration [31] was found at nearly 3450 cm^−1^ (Figure 1a,c,e), the N-H bending vibration [32] at nearly 1530~1500 cm^−1^ (Figure 1a,c), and the C=O [31] bond at approximately 1715 cm^−1^ (Figure 1a,c), demonstrating the isocyanate reaction with the -OH of PDF and the formation of carbamate in Al/PDF (29 μm and 13 μm). The characteristic peak at around 1100 cm^−1^ was caused by the C-O-C [28] in the main chain segment of the binder. The characteristic peaks located at nearly 1015 cm^−1^ and 900 cm^−1^ (Figure 1b,d) were due to N-F [28,33] bond vibrations. Meanwhile, the characteristic peak at nearly 800 cm^−1^ (Figure 1b,d) was attributed to the C-N [28,34] bond vibration, which verified that NF_2_ of Al/PDF with sizes of 29 μm and 13 μm was not damaged during the reaction. Notably, several Al/PDF peaks with a powder size of 1~3 μm (Figure 1e) were not evident, particularly the N-H bending vibration (1530~1500 cm^−1^), C=O bond characteristic peak (~800 cm^−1^), N-F bond characteristic peak (~1015 cm^−1^ and ~900 cm^−1^), and C-N bond characteristic peak (~800 cm^−1^). However, in contrast, a more substantial characteristic peak existed at approximately 965 cm^−1^, which is presumed to overlap with peaks occurring after the shift of several of the above characteristic groups. Nevertheless, this requires further characterization.

Furthermore, the morphology of Al powder and Al/PDF with different powder sizes was analyzed through FESEM. Figure 2 describes the morphologies. As described in Figure 2d–f, all Al/PDFs exhibited a spherical shape, with the PDF layer cladding stacked on the surface. The Al/PDF surfaces were smoother and flatter than the Al surfaces, which had rougher and sharper protrusions. Compared to Al/PDF with powder sizes of 29 μm and 13 μm, the 1~3 μm Al/PDF presented slight agglomeration. A typical mapping analysis was performed to investigate the distribution of PDF on the Al powder further. The element mapping results (Figure 2g–l) for Al/PDF and Al revealed the generation of the N and F elements after the coating, and these elements were evenly distributed on the Al sphere. By combining the images of the FTIR spectra, it could be verified that NF_2_ retained the original structure and was uniformly distributed on Al/PDF composites successfully. Figure 3 displays the TEM image of Al/PDF (1~3 μm). As indicated in Figure 3b,c, Al/PDF (1~3 μm) presented a typical core-shell structure, with the transparent film-like substance PDF adhering to the periphery of the Al powder. Based on the morphological characterization, it can be concluded that NF_2_ was distributed on the Al/PDF composites, and the thin coating layers of PDF formed core-shell structures. Additionally, the FTIR spectra of Al/PDF (1~3 μm) were validated and complemented by the TEM images.

Figure 4 depicts the XRD diffraction pattern of Al/PDF with different powder sizes. It was observed from Figure 4 that there was no significant change in the strong peak position of Al/PDF composites in comparison to the XRD standard spectra of Al powder. Besides, the angles of the five diffraction peaks in the three Al/PDF composites were maintained at 38.46°, 44.73°, 65.09°, 78.26°, and 82.45°. This finding aligned with the diffraction peaks of standard Al, and the corresponding diffraction crystal planes were (111), (200), (220), (311), and (222), respectively [1,7,14]. Notably, the crystal structure was a face-centered cubic structure, suggesting that the positions of the diffraction peaks were unchanged after adding PDF for Al powder.

Figure 5 demonstrates the XPS spectra of Al powder and Al/PDF composites of different powder sizes. These spectra were used to analyze the elements and states of the surface. The Al powder primarily consists of Al, O, and C, in which O and C elements result from the oxide layer. Correspondingly, more evident peaks of N and F elements can be observed in 29 μm Al/PDF, 13 μm Al/PDF, and 1~3 μm Al/PDF, which implies the presence of the PDF coating layer on the Al powder surface. The spectrum of C 1s (see Figure 5d–f) of Al/PDF displays signal peaks at ~284.4 eV, 284.8 eV, ~286.2 eV, ~286.6 eV, and ~289 eV positions after splitting, corresponding to C-H [5], C-C [5,7,9], C-O [5,9,35], C-N [33], and C=O [5,7,9,35] bonds, respectively. Meanwhile, the O 1s spectra (Figure 5g–i) of Al/PDF display signal peaks at 531.1~531.5 eV, 531.5~532 eV, and ~533 eV, which correspond to O-Al [7,36], O-C [36], and O=C [5,36] bonds, respectively. Combined with the analysis of the N element, the signal peak at 399.4 eV was -CO-NH- [19,33,37], confirming the occurrence of the curing reaction of PDF with the isocyanate groups of methylidynetri-p-phenylene tri-isocyanate on the surface of Al powder during the coating process. As depicted in the N 1s (~404.15 eV) [33,35] and F 1s (~686.55 eV) [35,38] spectra, element F still existed in the form of NF_2_, and the structure of the N-F bond was not damaged. Combining the other characterization results it could be observed that NF_2_ was not destroyed and routinely encapsulated through PDF on the surface of Al powder by curing with methylidynetri-p-phenylene tri-isocyanate.

### 3.2. Characteristics of Al/PDF Composites

Figure 6a–f illustrates the contact angles of water on the Al powder and Al/PDF composites with different powder sizes. Moreover, the specific values of these contact angles are listed in Figure 6g. Notably, after pressing and forming the flat plane, deionized water can quickly spread out the surface of Al powder, with the contact angle dropping as the particle size decreases. Meanwhile, the contact angle of deionized water on Al/PDF composites significantly increased by 51.24° (29 μm), 69.47° (13 μm), and 75.96° (1~3 μm). These results imply that Al/PDF composites generated greater hydrophobicity, thereby effectively protecting internal Al powder from moisture.

To provide NF_2_ for Al powder, an energetic fluorinated polymer PDF was bonded on Al powder. As exhibited in Figure 7, the actual amount of PDF layer can be roughly measured through PDF thermal decomposition [26,28,39] by TG analysis. The decomposition of PDF started at nearly 200 °C, leading to a mass loss of approximately 7.89 wt% (29 μm), 8.56 wt% (13 μm), and 9.29 wt% (1~3 μm) in Al/PDF. This phenomenon is attributed to the surface energy of Al powders with different powder sizes. As the powder size decreases, the specific surface area increases, enabling a more significant amount of PDF to be bonded on Al powder.

Figure 8 shows the ignition and combustion processes of the Al powder and Al/PDF composites of different powder sizes. Appendix A indicates the statistics of the ignition delay, and Appendix A displays the combustion duration. Figure 8 describes that the ignition delay of Al/PDF was reduced by NF_2_, with the ignition delay shorted 319 ms (29 μm), 177 ms (13 μm), and 157 ms (1~3 μm), respectively. Notably, the NF_2_ increased flame intensity and duration of combustion extended with 1708 ms (29 μm), 2005 ms (13 μm), and 1343 ms (1~3 μm), respectively. These findings imply that NF_2_ can assist in the combustion process of Al, thereby achieving the target of enhancing energy release. With decreasing powder size, the ignition delay of both Al and Al/PDF decreased together, the duration of combustion increased, and the combustion intensity improved. Furthermore, due to the significant role of NF_2_ through PDF, 1~3 μm Al/PDF was highlighted optimally. Facilitating minimum ignition delay and the most extended combustion duration.

### 3.3. The Catalytic Effect of Al/PDF on AP

Ammonium perchlorate (AP) is a common oxidant component in solid propellants. The burning rate of the solid propellant is closely related to the decomposition of AP [5]. Numerous studies have demonstrated that a reduction in the decomposition temperature of AP can increase the burning rate of the propellant. The catalytic effects of Al and Al/PDF samples of different powder sizes on the decomposition of AP were analyzed through DSC. Figure 9 reveals the DSC curves. As presented in Figure 9, the thermal decomposition of pure AP consists of the following three stages: an endothermic peak at 243.33 °C, which is attributed to the crystal transition, and two exothermic peaks at 298.50 °C and 403.12 °C, which correspond to the low-temperature and the high-temperature decomposition (THTD) of the AP exothermic process. Notably, all the Al-containing samples exhibit similar endothermic peaks as AP at nearly 243.5 °C. This indicates that the endothermic peak of AP is unaffected by the presence or absence of NF_2_ in PDF. It is observed that after adding Al-contained samples for AP, all low-temperature exothermic peaks of AP weaken and shift closer to high-temperature exothermic peaks. This implies that the exothermic process of AP changed from two separate phases to two consecutive phases, even nearly one phase. A more focused exothermic reaction favors further energy release between AP and Al in the solid propellant. Nevertheless, the peaks of THTD of AP catalyzed by Al are at 411.83 °C (29 μm), 399.67 °C (13 μm), and 405.50 °C (1~3 μm), respectively, in the absence of the catalytic effect of NF_2_ in PDF. The increase in the temperature of high-temperature exothermic peaks lowered the catalysis for AP. In contrast, THTD peaks of AP catalyzed by Al/PDF composites are located at 384.75 °C (29 μm), 368.52 °C (13 μm), and 356.47 °C (1~3 μm). Compared to the absence of NF_2_ of PDF, THTD was reduced by 27.08 °C (29 μm), 31.15 °C (13 μm), and 49.03 °C (1~3 μm), respectively. The Al/PDF composites can lower the decomposition temperature of AP more or less and promote the thermal decomposition of AP at high temperatures. Moreover, the catalytic effect of Al/PDF on AP increases with decreasing powder size.

According to the aforementioned DSC results and discussions, the main pyrolysis volatiles of mixed samples consisting of AP and Al/PDF composites (29 μm, 13 μm, and 1~3 μm) were analyzed through PY-GC/MS system at 300 °C and 500 °C. Chromatograms mixed samples pyrolysis are indicated in Appendix A. The majority of peaks were identified using the NIST library and previous studies [40,41]. The main compounds identified are listed in Appendix A. When the pyrolysis occurred at 300 °C, the pyrolysis products of Al/PDF and AP were detected to be dominated by elements C, O, and N, such as *N*-Methylethylenediamine (C_3_H_12_N_2_) and glyoxylic acid (C_2_H_2_O_3_), as well as some chlorine compounds such as 2-hexenedioic acid, 2,4-dichloro-5-oxo-(C_6_H_4_Cl_2_O_5_), ammonium chloride (NH_4_Cl), and chlorobenzene (C_6_H_5_Cl). With the decrease of the powder size, a small amount of silicone compounds exist in the pyrolysis products of Al/PDF and AP with 1~3 μm powder. Presumably due to the strong reactivity of Al/PDF at this powder size, which reacted with tubes containing Si during the pyrolysis reaction with AP, resulting in such as dodecamethylcyclohexasiloxane (C_12_H_36_O_6_Si_6_), octadecamethylcyclononasiloxane (C_18_H_54_O_9_Si_9_), tetradecamethylheptasiloxane (C_14_H_44_O_6_Si_7_), and other compounds. When the pyrolysis was at 500 °C, the main products were aminoacetic acid (C_2_H_5_NO_2_), alanin (C_3_H_7_NO_2_), and 3-amino-2-methylpropanoic acid (C_4_H_9_NO_2_). As the powder size decreased, some fluorine compounds and silicone compounds were detected. There is the generation of fluoro(trinitro)methane (CFN_3_O_6_) and dimethylphenylfluorosilane (C_8_H_11_FSi) in the pyrolysis of 13 μm Al/PDF with AP. Compounds such as dimethylphenylfluorosilane (C_8_H_11_FSi), fluoro(trinitro)methane (CFN_3_O_6_), and *N*-(Trifluoroacetyl)-*N*′-tetrakis (trimethylsilyl)norepinephrine (C_22_H_42_F_3_NO_4_Si_4_) were monitored from the pyrolysis products of AP and 1~3 μm Al/PDF. The organic volatiles generated from the pyrolysis of AP and Al/PDF at 300 °C and 500 °C are mainly compounds consisting of C, H, N, and O elements. Fluorine compounds and silicon compounds were more likely to be produced at 500 °C rather than at 300 °C. In the pyrolysis of AP and Al/PDF, the powder sizes of 13 μm and 1~3 μm Al/PDF pyrolysis the fluorine compounds with higher reactivity corroded the pyrolyzer tubes, leading to the production of silicon compounds.

Combined with the results and discussions of DSC and PY/GC-MS, the ignition and combustion tests were conducted on mixed samples of AP and Al to investigate the energy released from the combustion. Herein, the laser ignition and the combustion process of these mixed samples were recorded by the high-speed camera, and the images are recorded in Figure 10. The corresponding ignition delay and combustion duration data are depicted in Appendix A, respectively. Notably, the mixed sample of 29 μm Al has a long ignition delay. However, the mixed sample of Al/PDF with the same powder size shortened the ignition delay from 583 ms to 45 ms. During the combustion process, the combustion duration was nearly doubled (499 ms extended to 838 ms), and the intensity of the trailing flame during the combustion process was considerably increased. Moreover, the NF_2_ of Al/PDF enhanced the energy release during the combustion process with AP. Besides, the energy release of 13 μm Al/PDF and AP was promoted by NF_2_ during the combustion process with AP, with the ignition delay shorted by 134 ms and the combustion duration extended by 365 ms. Although the ignition delay between 1~3 μm Al and AP without NF_2_ (16 ms) was close to the sample of Al/PDF (15 ms), severe deflagration occurred during the combustion process, leading to an unstable combustion state with a duration of only 266 ms. In contrast, the stability of 1~3 μm Al/PDF and AP was significantly enhanced without any deflagration in the presence of NF_2_ of PDF, and the ignition delay remained short. Thus, the stability of 1~3 μm Al/PDF and AP was drastically enhanced without any deflagration phenomenon. Furthermore, the intensity of the tail flame is considerable, and the combustion duration is extended to 1153 ms. With the powder size of Al/PDF composites decreased, the ignition delay of the mixed samples was shortened, the burning time was increased, and the intensity of the tail flame was increased.

In order to further understand the reaction processes and changes after ignition, the combustion residues of Al (Al/PDF) and AP adhering to the walls of the quartz tubes were also investigated. These burned-out particles were collected and analyzed by SEM and EDS. The graph of SEM is exhibited in Figure 11, and the results of EDS are demonstrated in Appendix A. According to previous studies [42,43], the powder size of Al has a significant decisive influence on Al combustion characteristics. When large agglomerates are generated, combustion efficiency will be reduced. From Figure 11, it can be seen that the combustion of Al with AP after ignition in the case where there is no NF_2_ produced agglomerates with the same or even larger particle size than the original ones, and with the increase of the particle size, there is still some unreacted Al powder in the residue, which also corresponds to the phenomenon in Figure 11a. All of the above indicates the incomplete oxidation of Al powder during the combustion reaction. According to the results of the characterization of EDS, the main components of the above residues are Al_2_O_3_, Al powder incomplete with oxidation, and C-containing particles. After ignition, Al/PDF with different powder sizes provided NF_2_ to the combustion reactions with AP, and the residues produced in these reactions changed significantly compared to Al without NF_2_. Although agglomeration of the residues still occurred, the particle sizes were significantly reduced compared to the state when no ignition occurred. This implies that NF_2_ of Al/PDF improved the combustion and reaction efficiencies with AP, which corresponds to the combustion ignition diagrams of Figure 11. This is mainly due to the high reactivity of NF_2_ of Al/PDF, which reacts violently with AP and Al on the surface, respectively. The high temperature generated during the combustion reaction induces the Al vapor to escape from the interior of Al and immediately react with the oxygen in the air to form nanoscale Al_2_O_3_ smoke particles that condense and attach to each other and finally form agglomeration (Figure 11d–f).

## 4. Conclusions

In summary, the Al/PDF composites with different powder sizes were efficiently synthesized through the solvent/non-solvent method. The corresponding characterization demonstrated that NF_2_ with no damage and the PDF content of Al/PDF surface was found to be 7.89 wt% (29 μm), 8.56 wt% (13 μm), and 9.29 wt% (1~3 μm), respectively. Al/PDF has superior hydrophobicity than Al, with the static contact angle increasing by 51.24° (29 μm), 69.47° (13 μm), and 75.96° (1~3 μm). The NF_2_ effect of energy release in Al/PDF was studied. The ignition delay of Al/PDF was shortened, and Al/PDF presented more excellent catalytic behavior than Al on AP in DSC curves. The joining of NF_2_ in the ignition and combustion of Al/PDF and AP improved the combustion efficiency of Al by proving PY-GC/MS and combustion residues, which detected fluorinated gas compounds and residue particle sizes that were significantly reduced. More impressively, NF_2_ of Al/PDF with a powder size of 1~3 μm presented impressive catalytic behavior on AP; the ignition delay and the combustion duration were 15 ms and 1153 ms, respectively. Most importantly, this study highlights the positive impact of NF_2_ of Al/PDF composites, prepared through the rapid and efficient method, in enhancing Al energy release, with the ignition and combustion properties of Al/PDF improving significantly.

## Data Availability

The original contributions presented in this study are included in the article/Appendix A. Further inquiries can be directed to the corresponding author.

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
