# Peer review of "Miraculous Al/PDF Composites Using NF2 to Enhance the Energy Release of Al, Prepared Through an Efficient Method"

_nanomaterials, 2024, doi:10.3390/nano14241980_

Round 1

Reviewer 1 Report

Comments and Suggestions for Authors

The paper is very interesting, contains a lot of new and valuable information.

Author Response

Thank you very much for taking the time to review this manuscript and acknowledgement of the study of our research. We also wish the reviewers well in their research studies!

Reviewer 2 Report

Comments and Suggestions for Authors

The article is a judicious work carried out in a similar way to the one presented previously for Boron, with its own conclusions obviously.

Although it is not bad, the authors practically saved a script from the previous article to replicate the results of Al in this new article.

The figures in general should be reviewed for size, some are very small.

Author Response

Thank you very much for taking the time to review this manuscript and for your suggestions. We have responded to the suggestions raised, thank you again for taking the time to review the revised manuscript.

Al and Boron (B), are metal fuels used in solid propellants, we have studied them accordingly. Previously the enhancement of the properties of B were studied and B/NF2 composites were prepared successfully. Afterwards, the study of the enhancement of the properties of Al was carried out by similar preparation and characterization methods, which led to the similarities between this manuscript and the earlier article of B. The relevant explanation has been added in the Introduction. In addition to the direct improvement of ignition and combustion performance, the hydrophobicity of Al/NF2 composites plays an important protective role compared to B/NF2. The dimensions of some of the figures have been modified.

Reviewer 3 Report

Comments and Suggestions for Authors

For their study, the authors prepare Al/PDF composites through surfactants and the solvent/non-solvent method, to enhance the energy release of Al powder in solid propellant. There is a good introduction to the subject and review of the antecedents. Regarding the description of the experimental part and the work performed here, this is clear and understandable. On the other hand, there are a good presentation, discussion and analyses of the obtained results. Which made it possible to write comprehensive conclusions. The manuscript has the potential to be published, but first I would like the authors to clarify some aspects of their work.

To authors

1.      In what type of engine or reactor could this fuel be used??

2.      Regardless of the potential of the composite as a fuel, what can you comment on the method and materials used to obtain the compost, I mean is it sustainable?

3.      How hazardous are the products of combustion to the environment and living beings?

4.      How corrosive is the fuel and combustion products, considering that there is fluorine at all times and generation of hydrofluoric acid?

5.      How expensive is this type of fuel compared to those already on the market?

6.      do you consider it important to measure the surface area of the Al/PDF powder and the influence this may have on combustion? If so, why was it not measured?

7.      Would you can explain why the hydrophobicity of the powder is important.

8.      Figure 6g shows that the smaller the powder, the less hydrophobic it is. Is this good or bad for the ignition and combustion of the fuel?

Author Response

Thank you very much for taking the time to review this manuscript and for your questions and suggestions. We have responded to the questions raised, thank you again for taking the time to review the revised manuscript.

Comments 1: In what type of engine or reactor could this fuel be used?

Response 1: Al/PDF prepared in this manuscript was used as metal fuel in solid propellants, to improve Al ignition delay and combustion efficiency, thereby enhancing the energy release.

Comments 2: Regardless of the potential of the composite as a fuel, what can you comment on the method and materials used to obtain the compost, I mean is it sustainable?

Response 2: No toxic or hazardous gases are generated during the entire preparation process, and the chemical reagents used and recycled are commonly used, as the composites for solid propellant, the process of preparation is sustainable.

Comments 3: How hazardous are the products of combustion to the environment and living beings?

Response 3: As the solid propellant material, most of the time only combustion-based reactions occur in the experimental environment, enabling good ventilation and disposal of reaction residues. According to the PY/GC-MS and residue analyses in the manuscript, there is no obvious generation of toxic or hazardous substances to the environment.

Comments 4: How corrosive is the fuel and combustion products, considering that there is fluorine at all times and generation of hydrofluoric acid?

Response 4: Before ignition, F exists as a polymer with no apparent corrosive properties. During combustion most of the F reacts with C, H, N, and O and combines and volatilises as a gas, the stronger reactivity of the F may react with the propellant shell at high temperatures, and the rest of the F reacts with the metal component mode. At the end of combustion F exists as a stable compound of the compound with no significant corrosive properties.

Comments 5: How expensive is this type of fuel compared to those already on the market?

Response 5: The cost of the synthesized composites is mainly focused on the cost of the raw materials used, with significant savings in labour costs due to the quick and easy preparation steps. Based on the results of the properties characterization, the economic input is met.

Comments 6: Do you consider it important to measure the surface area of the Al/PDF powder and the influence this may have on combustion? If so, why was it not measured?

Response 6: The surface area is related to the powder size, and the manuscript has clearly used three different powder sizes of Al. Causes composite material properties enhancement is PDF, no special needs there is no need to test the surface area.

Comments 7: Would you can explain why the hydrophobicity of the powder is important.

Response 7: Al is reactivity, easy to react with oxygen and water in the air, resulting in the generation of the oxide layer dominated by Al2O3 on the surface of Al, and causes Al to absorb moisture and agglomerate, ultimately leading to the difficult ignition and short duration of combustion shown in Figure 8 (a,c, e) and Figure 10 (a, c, e).

Comments 8: Figure 6g shows that the smaller the powder, the less hydrophobic it is. Is this good or bad for the ignition and combustion of the fuel?

Response 8: The decrease in Al powder size is accompanied by an increase in surface area, which means an increase in reactivity, easier ignition and easier contact with water. For raw Al, the reduction in powder size makes poor hydrophobicity, which can seriously affect the ignition delay and combustion efficiency of Al. For Al/PDF composites, the hydrophobicity of the composite itself is greatly improved, and the reduction in hydrophobicity due to the reduction in powder size is much smaller than that of raw Al, which is acceptable according to the results of properties characterization.

Reviewer 4 Report

Comments and Suggestions for Authors

Dear authors, 

Thanks very much for your interesting manuscript dealing with the manipulation of the energetic reactivity of Al powder as solid propellant, through minimizing the natural formation of aluminum oxide and incorporating reactive linking components in its surface.

In this work Al/PDF-NF2 composites were synthetized using three different Al particle sizes which were pretreated in NaOH solutions and reacted with specific surfactants under the solvent/non-solvent method. 

The prepared Al/PDF were characterized in its surface morphology by FTIR, FESEM, DRX; in its physico-chemical attributes by contact angle measurement in water,  thermogravimetric analysis and differential scanning calorimetry complemented with high-speed camera inspection of the combustion process and combustion residue characterization

This manuscript is well written with a rational sequence of ideas that makes an easy reading.  The discussion is appropriate and logical.  Unfortunately, there are some sentences and inappropriate words that need to be revised. These are yellow highlighted in the attached manuscript with additional pop-up messages. 

Also, the references are appropriate to facilitate readers a rapid enlightenment in this investigation.  

  My recommendation is to publish after correcting the manuscript.

Author Response

Thank you very much for taking the time to review this manuscript and for your questions and suggestions. We have amended and added to the suggested areas accordingly, please refer to the content of the PDF file for detailed responses. Sections that have been revised in the Word manuscript have been highlighted in red. Thank you again for taking the time to review the revised manuscript.
